# Bioinformatics Prediction for Network-Based Integrative Multi-Omics Expression Data Analysis in Hirschsprung Disease

**DOI:** 10.3390/biom14020164

**Published:** 2024-01-30

**Authors:** Helena Lucena-Padros, Nereida Bravo-Gil, Cristina Tous, Elena Rojano, Pedro Seoane-Zonjic, Raquel María Fernández, Juan A. G. Ranea, Guillermo Antiñolo, Salud Borrego

**Affiliations:** 1Department of Maternofetal Medicine, Genetics and Reproduction, Institute of Biomedicine of Seville, University Hospital Virgen del Rocío/CSIC/University of Seville, 41013 Seville, Spain; 2Center for Biomedical Network Research on Rare Diseases (CIBERER), 41013 Seville, Spain; 3Department of Molecular Biology and Biochemistry, University of Malaga, 29010 Malaga, Spain; 4Biomedical Research Institute of Malaga, IBIMA, 29010 Malaga, Spain; 5Center for Biomedical Network Research on Rare Diseases (CIBERER), 29071 Malaga, Spain; 6Spanish National Bioinformatics Institute (INB/ELIXIR-ES), Instituto de Salud Carlos III (ISCIII), 28029 Madrid, Spain

**Keywords:** Hirschsprung’s disease, enteric neuropathy, system biology, omics expression data, networks analysis

## Abstract

Hirschsprung’s disease (HSCR) is a rare developmental disorder in which enteric ganglia are missing along a portion of the intestine. HSCR has a complex inheritance, with *RET* as the major disease-causing gene. However, the pathogenesis of HSCR is still not completely understood. Therefore, we applied a computational approach based on multi-omics network characterization and clustering analysis for HSCR-related gene/miRNA identification and biomarker discovery. Protein–protein interaction (PPI) and miRNA–target interaction (MTI) networks were analyzed by DPClusO and BiClusO, respectively, and finally, the biomarker potential of miRNAs was computationally screened by miRNA-BD. In this study, a total of 55 significant gene–disease modules were identified, allowing us to propose 178 new HSCR candidate genes and two biological pathways. Moreover, we identified 12 key miRNAs with biomarker potential among 137 predicted HSCR-associated miRNAs. Functional analysis of new candidates showed that enrichment terms related to gene ontology (GO) and pathways were associated with HSCR. In conclusion, this approach has allowed us to decipher new clues of the etiopathogenesis of HSCR, although molecular experiments are further needed for clinical validations.

## 1. Introduction

Hirschsprung’s disease (HSCR) is a neurocristopathy, caused by defective migration, proliferation, differentiation and/or survival of neural crest cells, leading to gut aganglionosis. It is a rare congenital disease, with an incidence of 1 in 3500–5000 live births [1]. The disease is characterized by high heritability (>80%), significant sex bias (male: female ratio, 4:1) and high sibling and recurrence risk (3 to 17%) [2]. The combination of all these distinctive characteristics are the hallmark of a typical multifactorial genetic disorder [3]. Genetic studies on HSCR have identified more than 20 disease-associated genes, mainly *RET* (ret proto-oncogene) and *EDNRB* (endothelin receptor type B) [1]. However, all these genes cumulatively explain <10% of cases, suggesting that additional genes may also be involved in the etiology of HSCR [3,4].

On the other hand, increasing evidence indicates that numerous microRNAs (miR-NAs) are altered in HSCR [4,5,6]. MiRNAs are a class of small noncoding RNAs with function in posttranscriptional regulation of gene expression, thereby modulating diverse biological processes, such as neuronal cell differentiation, development, plasticity and survival [7,8]. These regulators are involved in modulating the pathogenesis of HSCR by directly suppressing a variety of functional targets [4,5,6]. MiRNAs have also been observed to be dysregulated in a variety of human pathologies, including cancer, neurodegenerative diseases and cardiovascular diseases [9,10,11]. The ability of chosen miRNAs to target various disease-associated mRNAs makes them interesting candidates as targets of therapeutics (in the form of anti-miRNAs) or as therapeutics (in the form of miRNA mimics) [12].

A common challenge in rare disease research is to identify and understand the cellular and molecular mechanisms underlying the pathophysiology of the disease. The knowledge underlying normal gut development and motility by identification of dysregulated pathways in HSCR and their genetic causes may provide new possible treatment strategies. Over the past decades, many different approaches have been applied to human genetic studies—beginning with classical positional cloning, linkage analyses and genome-wide association studies (GWAS)—to the recent next-generation sequencing (NGS) studies, which have allowed us to identify novel genetic variants and genes linked to HSCR [13,14,15,16,17].

Studies implementing large-scale techniques (omics technologies) have increased our understanding of disease mechanisms and led to the discovery of new biological pathways, genetic loci underpinning disease progression, biomarkers, and targets for therapeutic development [18,19,20,21]. Based on the hypothesis that molecular features within a system establish functional connections or are part of modules to carry out biological processes [21], advances in omics technologies allow for holistic studies into biological systems [21]. These approaches further can help us to identify possible connections (e.g., genotype–phenotype correlations) and/or subnetworks (e.g., biological pathways) that are informative of an observed phenotype [22]. Biological systems can be abstracted as a series of networks of interconnected molecular entities (genes, proteins, lncRNAs, miRNAs, etc.) [23], whereas a network is a representation of relationships (edges) between entities (nodes) [24]. In general, network-based approaches obtain disease-related information from complex topological patterns [24]. Networks commonly used in systems biology include gene regulatory networks, protein–protein interaction (PPI) networks, literature-curated networks and hybrid networks [25].

This study aims to identify new HSCR-associated modules, genes and miRNAs by applying a designed workflow based on a system biology approach, which integrates multi-omics information and combines different biological networks to illustrate associations among multiple molecular features and to discover new biomarkers for this disease.

## 2. Materials and Methods

An overview of the different methods applied in this study is illustrated in Figure 1. In summary, a total of four major steps have been executed as follows: (1°) collecting and preprocessing expression data; (2°) identification of likely disease-associated modules and new disease candidate genes; (3°) prediction of disease-associated miRNAs; (4°) miRNA biomarker identification. Moreover, we included two additional working packages: (5°) functional enrichment analysis and (6°) disease relevance evaluation, which were conducted after steps 1, 2 and/or 3, as depicted in Figure 1.

### 2.1. Data Collection

Available omics expression data of HSCR patients were searched in Omics Discovery Index (OmicsDI—www.omicsdi.org (accessed on 2 January 2023) [26,27]. The dataset inclusion criteria for the analysis were as follows: (1) presence of data on mRNA, miRNA and protein expression in colon tissue samples of HSCR patients; and (2) possibility of performing comparison groups of aganglionic and ganglionic segments of colon tissue samples. These requirements led to the following sets being selected for analysis: GSE98502 [28], GSE96854 [29], GSE77296 [30] and PXD021292 [31]. A summary of these four individual studies is shown in Appendix A.

Moreover, HSCR-related genes were collected from four databases as follows: the Comparative Toxicogenomics Database (CTD) [32], DisGeNET [33], HuGENet [34] and Malacards [35,36] (Appendix A). HSCR-associated miRNAs were also extracted from these four databases together with HMDD v3.0 [37] (Appendix A). The protein–protein interaction (PPI) data were obtained from the HIPPE database [38]. MiRNA–target interaction (MTI) data were collected from three different online databases as follows: DIANA [39], miRecords [40] and miRTarbase v.8 [41].

### 2.2. Data Preprocessing and Differential Expression Analysis

Selected microarray datasets (GSE98502 [28], GSE96854 [29] and GSE77296 [30]) of HSCR patients in the GEO database [42] were individually investigated for differential expression (DE) patterns between ganglionic and aganglionic colon samples by using the same analytical tool, GEO2R [43] (GEO’s online tool for analyzing GEO data, available at http://www.nci.nlm.nih.gov/geo/geo2r/ (accessed on 2 January 2023)), to maintain consistency during individual analysis. To perform differential expression (DE) analysis, we selected Force normalization in the Options tab, the false discovery rate (FDR) *p*-value adjustment for multiple testing by Benjamini–Hochberg method [44], the log data transformation method to normalize the results and the cutoff value to filter out the results of FDR ≤ 0.05 and |logFC| ≥ 0.5.

GEO2R [43] provided a list of probes and corresponding gene/miRNA symbols ranked according to their degree of DE. In the second step, the gene/miRNA symbol annotation of probe sets was updated. If a particular gene/miRNA was mapped to several probe sets, the highest expression value was selected for further analysis, while probes without gene/miRNA annotations were removed.

On the other hand, the list of significant DE proteins (with a |fold change| ≥ 1.2 and a *p*-value ≤ 0.05) between the ganglionic and aganglionic regions of HSCR patients, which were in common among the male L-HSCR, male S-HSCR, and female S-HSCR assays, was downloaded from the originally published article [31]. The “Retrieve/ID Mapping” tool of UniProt [45] was used to retrieve UniProt Knowledgebase (UniProtKB) proteins from the list of DE protein identifiers and to convert Uniprot protein ID into respective gene symbols.

Finally, the multi-symbol checker tool [46] was used to check if the gene nomenclature was HGNC-approved symbols. Otherwise, the miRBase miRNA-version and miRNA-to-precursor converter options, from the miRNA Enrichment Analysis and Annotation Tool (miEAA) miEAA 2.0 [47], were used to support research settings where older releases of miRBase [48] are in use.

### 2.3. Identification of Novel Candidate Disease Genes and Disease-Associated Modules

We employed a previously described method [49] for predicting disease risk genes, based on currently known disease-associated genes collected from DisGeNET [33] and differentially expressed genes according to omics expression data and PPI network analysis.

In the first step, we created the HSCR disease-relevant PPI network by selecting data from the Human Integrated Protein–Protein Interaction reference (HIPPIE) database [38] and determined high-density clusters in the PPI network according to DPClusO algorithm by using nine different and independent density values from 0.1 to 0.9 [49,50,51,52,53,54].

In a second step, cluster enrichment of DE disease-associated genes was determined by Fisher’s exact test, and a significance score (SScore) [49], a measure of confidence of prediction of each gene based on the *p*-values of the clusters they belong to, was assigned to each gene.

In a third step, receiver operating characteristic (ROC) analysis [55,56] was used to evaluate the relevance of SScore [49] to predict DE disease-associated genes collected from DisGeNET [33] as well as the optimal cluster density value to discriminate among them.
(1)SScore=−log(p‐value)

Therefore, clustering stratification outcome (total number of clusters) and cluster membership (cluster composition) were optimized by identifying the cluster density which allow obtain the best HSCR disease-associated gene prediction. In this light, we searched the optimal “outcome-driven” stratification, measured by the known good-quality HSCR gene distribution across clusters obtained (based on area under the ROC curve values for outcome classification).

Finally, the statistically significant clusters (disease-associated modules), and the potential new disease-associated genes (comprised in these clusters) were determined after multiple testing correction by Benjamini–Hochberg method [44].

### 2.4. Prediction of Disease-Associated miRNAs

We applied a previously reported method [57] for predicting disease-associated miRNAs based on currently known disease genes. In summary, we built a set of genes related to HSCR by combining the reported HSCR disease-associated genes in the four databases previously mentioned [32,33,34,35,36] (Appendix A) together with the set of potential candidate disease genes identified in this study. This list was used as a query to find their miRNA targets and to create the MTI datasets. This information was collected from the three different validated online databases of MTI [39,40,41,58] with the same selection criteria described in the original method [57].

The interactions between miRNAs and gene targets obtained from each of the three interaction sources were individually represented as a bipartite graph, which is called MTI network. The biclusters (miRNA-regulatory modules, MRM) of these networks were determined by BiClusO algorithm [54,59,60]. Then, we created HSCR-related sub-MRMs from the MRMs obtained by identifying the presence of HSCR genes. Finally, we assigned a relevance score (RS) [57] for each miRNA present in these sub-MRMs. The relevance score of ith miRNA (RS_miRNA(i)_) is given by
(2)$${RS}_{miRNA(i)}={NoofHSCR}_{miRNA(i)}\times {Noofcluster}_{miRNA(i)}$$
where NoofHSCR_miRNA(i)_ is the number of HSCR genes attached to ith miRNA in the HSCR MRM set and Noofcluster_miRNA(i)_ is the number of HSCR MRMs attached to ith miRNA.

After normalizing the score of individual miRNAs in each dataset, we calculated the total relevance score (TSR) [57] for each miRNA as follows:(3)$${TRS}_{miRNA(i)}=\sum_{n=1}^{3}\frac{RSni}{Cni}\sum_{n=1}^{3}Eni$$
where TRS_miRNAi_ is the total relevance score of ith miRNA based on all datasets, RSni is the relevance score of ith miRNA in nth dataset, Cni the number of clusters in nth dataset and Eni is the Boolean value, measuring whether ith miRNA is in the nth dataset.

Every miRNA with a total relevance score (TSR) > 0 was proposed in this study as a candidate disease-associated miRNA.

### 2.5. Identification of Potential miRNA Biomarkers in HSCR

To reveal potential candidate miRNA biomarkers in HSCR, we used the bioinformatics tool miRNA-BD [61]. In this tool, the biomarkers are identified based on two parameters: the values of single-line regulation (NSR) and transcription factor percentage (TFP) on the condition-specific MTI network.

According to the user guide of this tool, miRNAs from the input data with significantly high NSR and TFP values (cutoff *p*-value  ≤  0.05 in Wilcoxon signed-rank test), calculated based on the disease-specific MTI network, were selected as a candidate disease biomarker. Moreover, we provided the disease-specific MTI networks, TF gene datasets and different kinds of input datasets (list of DE and/or disease-associated miRNA/mRNAs).

Thus, we identified potential miRNA biomarkers from (a) the list of predicted HSCR-associated miRNAs obtained in this study and (b) the list of DE miRNAs obtained from GSE77296 [30].

The HSCR-specific MTI network was constructed in the following way: (a) extracting from miRNet 2.0 [62] those target genes of DE miRNAs in HSCR and HSCR-known miRNAs according to three different databases (DIANA [39], miRecords [40] and miRTarBase [41]); and (b) extracting from miRNet 2.0 [62] the target miRNAs of DE genes in HSCR, known HSCR-associated genes and proposed new candidate HSCR genes obtained in this study. Finally, the union of all these miRNA–target interaction pairs was selected to create the MTI network by the web-based tool OmicsNet 2.0 [63]. NAP [64] was employed for visual analysis of the biological network.

Moreover, TF gene dataset was obtained from TRRUST v.2 [65], the reference database of human transcriptional regulatory interactions.

### 2.6. Functional Enrichment Analysis

Malachite [66], a Gene Enrichment Meta-Analysis (GEM) Tool for ToppGene [67], was used twice, to analyze and compare differences at once between (1) the DE gene datasets and (2) the statistically significant disease-associated modules identified. The enrichment categories tested were drugs, diseases, pathways and gene ontology terms. Moreover, the same enrichment categories were analyzed for the new candidate disease genes using ToppFun [67]. Finally, the functional enrichment analysis of predicted miRNA was performed with miEAA 2.0 [47].

HSCR-associated GO terms and pathways were filtered out from CTD [32] and Malacards [35,36] using the keyword “Hirschsprung”.

### 2.7. Disease Relevance Evaluation

The implication of candidate HSCR-related genes and miRNAs in enteric nervous system (ENS) and/or HSCR onset was assessed by performing an automated bibliography search using a previously published script [15]. Moreover, the importance of the proposed HSCR-related genes in biological systems was explored by searching several databases [65,68,69,70,71,72]. The total number of HSCR-related genes attached to new predicted HSCR-related miRNAs in the MTI networks created for this study was determined. Predicted miRNAs interacting with the clinical gene panel for HSCR in ClinGen [73] were highlighted. TAM 2.0 database [74] was employed to identify specific colon tissue miRNAs among the predicted HSCR-related miRNAs.

Predicted miRNAs with altered expression in HSCR stenotic colon tissue was further studied. For this purpose, the HSCR miRNA expression profiles from the publicly available microarray dataset GSE77296 [30] was re-analyzed by GEO2R [43] and compared with our results. Moreover, the list of predicted miRNAs was compared with reported DE miRNAs obtained from the GSE77296 dataset [30] by other citing articles [30,75,76,77]. Additionally, we performed a manual search of altered expression in HSCR colon tissue [4,5,78,79,80,81,82,83,84] for the predicted miRNAs previously associated with HSCR. On the other hand, a manual review of miRNAs previously described as potential peripheral HSCR biomarkers [5,75,81,84,85] was conducted. The miRNA disease network was created with miRNet [62] using as a query the list of predicted HSCR-related miRNAs to check if HSCR is present in it.

Finally, we also studied if the predicted miRNAs target circular RNA (circRNAs, a type of single-stranded RNA, which, unlike linear RNA, forms a covalently closed continuous loop) differentially expressed between diseased tissue and paired normal intestinal tissues from patients with HSCR [86,87,88,89] by CircInteractome [90]. In this line, predicted miRNAs with interaction with HSCR-related lncRNAs were identified, for which a miRNA-lncRNA network was constructed in miRNet [62] using as query the list of 13 HSCR-related lncRNA (obtained by searching in four databases: CTD [32], DisGeNET [33], HuGENet [34] and Malacards [35,36]), to know *AFAP1-AS1*, *EDNRB-AS1*, *FALEC*, *HOTTIP*, *IQCF5-AS1*, *LCT-AS1*, *LINC00327*, *LINC01518*, *LINC01844*, *LOC105378311*, *MEG3*, *MIR31HG* and *THBS4-AS1*.

### 2.8. Statistical Analysis

We analyzed data statistically using SPSS 20.0 (IBM, Chicago, IL, USA). False discovery rate (FDR) correction by Benjamini–Hochberg method [44] of the uncorrected *p*-values was performed using the SDM FDR online calculator [91]. The VENN DIAGRAMS tool [92] was used for comparing gene/miRNA lists.

## 3. Results

### 3.1. Identification of DE Genes in HSCR and Gene Enrichment Meta-Analysis

The analysis of the two mRNA microarray datasets by GEO2R [39] allowed for the identification of 89 DE genes in GSE98502 (8 upregulated and 81 downregulated genes) and, 476 DE genes in GSE96854 (220 upregulated and 256 downregulated genes).

Moreover, we included a list of 651 DE proteins (coming from a total of 666 protein-coding genes) which were common among the male L-HSCR, male S-HSCR, and female S-HSCR assays previously published [31]. This list included 48 DE proteins upregulated, 58 downregulated, and 545 with different tend of regulation based on HSCR phenotype and/or patient sex.

The Venn diagram of the overlapping DE genes between these three datasets is shown in Figure 2a. We took the union set of the DE genes from these three comparisons and combined these DE genes into a single set consisting of 1180 genes. We found that 31 of the DE genes were reported in the DisGeNET database as HSCR-related genes. We considered these 31 genes as “known HSCR DE genes” (HDEGs) and the other 1149 DE genes as “other DE genes” (ODEGs).

Interestingly, the DE genes are slightly overlapped (3.98%) in these three datasets. Moreover, overlapped genes are differentially expressed in the same direction of regulation across all datasets except for *APOC3*, *GC* and *APOA1*, which were downregulated at the mRNA level in GSE96854 but upregulated at the protein level in PXD021292 [31] of aganglionic bowel compared with ganglionic segments of colon samples. Figure 2b shows the complete details of the list of overlapped DE genes found in the three datasets.

We applied Malachite to these three datasets to conduct enrichment analyses on these HSCR omics expression data. The significant GO terms for “Biological Process” and “Cellular component” enriched in all three datasets are listed in Appendix A.

Although no pathway was statistically significant enriched in all three datasets, we found that most GO terms’ enrichment was implicated in nervous system developmental processes, namely axon, synapse, cell adhesion and neuron projection development, HSCR-associated GO terms. Moreover, we found 4 diseases and 29 drugs or small molecules also enriched in all three datasets (Appendix A). All shared diseases presented neurological symptoms, while several of the small molecule compounds enriched in common were shown to alter the behavior of neurons, such as acetylcholine, dopamine, glutamate, GABA and serotonin, and/or there were substances involved in muscle contraction, such as Lant-6 and Lupex (Appendix A).

Therefore, despite the limited number of common DE genes obtained from these three omics experiments, the number of significant shared enrichment terms in all three datasets is higher and comprises a bigger number of DE genes.

### 3.2. Identification and Enrichment Analysis of Candidate Disease-Associated Modules and Potential New Disease Genes

From the 1180 DE genes previously identified, we constructed the relevant HSCR PPI network based on the HIPPIE database. A total of 14,343 interactions were collected involving 3806 different proteins from 3850 protein-coding genes (29 HDEGs, 988 ODEGs and 2833 interacting genes named “other genes, OG”).

The main topological characteristics of this PPI network obtained with NAP are shown in Table 1. Of note, this network is typically scale-free (fit power law TRUE), which gives our PPI network some important features such as stability, invariability to changes of scale and vulnerability to targeted attack. Network’s diameter is large, with more than six-step separation, which means that proteins are not very closely linked in spite of the average path length being 3.83. The transitivity or clustering coefficient of our PPI network is low, showing that the network contains few communities or groups of nodes that are densely connected internally, which is also reflected by the low value of average number of neighbors.

On the other hand, Table 2 provides the results of each DPClusO clustering and Figure 3 shows the area under the curve (AUC) for the ROC curves performed for each cluster density. We observed that a smaller density value resulted in a larger size and fewer number of clusters (Table 2).

The area under the ROC curves (AUC) described a poor discrimination with values in the range of 0.5 < AUC < 0.7, which may be due to incomplete information on known good-quality HSCR genes (Appendix A). The highest AUC (0.62) was obtained in the case of the cluster set generated using density = 0.5, for which this cluster dataset was selected to identify the candidate disease-associated modules and predict the new potential HSCR genes. A total of 55 out of the 563 initial statistically significant clusters were finally proposed as candidate disease-associated modules after adjusting the *p*-value for multiple testing (Appendix A). These 55 clusters comprised a total of 195 different genes (17 HDEGs, 59 ODEGs and 119 OGs). Therefore, after excluding the 17 HDEGs, these 178 other genes (with adjusted *p*-value < 0.05) were proposed as new HSCR candidate genes (Appendix A).

To validate our results, we searched how many of the predicted genes were exactly matched with reported HSCR genes by four reliable sources (Appendix A).

After considering overlapping between databases [32,33,34,35,36], we found that 14 out of 178 purpose genes (7.87%) matched with reported HSCR genes. Bearing in mind that our predictions are based only on a limited set of reported HSCR genes (a total of 527; Appendix A), the 7.87% coincidence with good-quality data is significant (*p*-value = 0.00027, calculated based on a hypergeometric distribution assuming a total number of human genes of 20,000). Even if we consider as universal just the total number of protein-coding genes in our PPI network (3850) and the set of included reported HSCR genes (196), the number of successes that HSCR genes found among our list of candidate HSCR genes is statistically significant (0.031). Moreover, after manually reviewing the results obtained from the automated bibliography search of the implication of these genes in enteric nervous system (ENS) and/or HSCR onset, we found that 39 of these 178 genes were associated with the ENS and/or HSCR development (Appendix A). As expected, the 14 predicted HSCR-related genes that matched with reported HSCR genes in any of the four gene–disease databases (Appendix A) have been previously reported as implicated in the enteric nervous system (ENS) and/or HSCR onset.

Furthermore, we found that 25 of our candidate HSCR-related genes are described as essential, 17 as oncogenes, 25 as tumor suppressor genes, 44 as housekeeping genes and 8 as transcription factors (Appendix A).

Gene enrichment analysis for all 178 genes was performed with ToppFun. The top 10 significant enrichment GO terms and pathways are shown in Table 3.

According to the values observed both in the pathways and in the enriched GO terms, approximately 30% of the 178 candidate genes are related to cell adhesion processes (Table 3). Other cellular processes regulated by cell adhesion such as cell morphogenesis and tissue development in multicellular organisms were also found in the top 10 enriched GO: biological process terms (Table 3). Moreover, HSCR-related enriched terms were found both in the molecular function category (signaling receptor binding) and in the cellular component category (membrane raft and plasma membrane region) (Table 3). We also found several HSCR-related pathways among the Top10 most significant pathways (Table 3), for example, signaling events regulated by Ret tyrosine kinase obtained from Malacards [35,36], and developmental biology and signaling by interleukins obtained by filtering out from CTD using “Hirschsprung” as a keyword.

Cluster enrichment analysis for the 55 potential disease modules was performed with Malachite in a similar way to that which we mentioned above. REACTOME pathway analysis showed that several pathways related to signal transduction, such as the MAPK signaling pathway and Ras signaling pathway, were detected across more than 50% of potential disease modules (Figure 4). In this line, KEGG pathway analysis showed that the VEGF signaling pathway was also detected across more than 50% of potential disease modules.

Other HSCR-associated pathways (Figure 4) obtained by filtering out from CTD and Malacards were central carbon metabolism in cancer, thyroid cancer; B cell receptor signaling, signaling by ERBB4, GRB2 events in ERBB2 signaling, SHC1 events in ERBB2 signaling, SHC1 events in ERBB4 signaling and the Fc epsilon RI signaling pathway.

Finally, we proposed two new HSCR-associated pathways, since they were observed in more than 50% of potential disease modules (Figure 4). The proposed new HSCR-associated pathways were the prolactin signaling pathway and aldosterone-regulated sodium reabsorption (Figure 4b).

The only enrichment GO term for “Biological Process” found in more than 50% of significant clusters was peptidyl-tyrosine autophosphorylation (GO:0038083). The wide functional diversity of disease modules is reflected by the high number of significant singleton enrichment GO terms and pathways, which was also remarkable. In this line, we found more than 2371 different and unique GO (BP) terms. As expected, among the enrichment diseases, we found HSCR (C0019569) present in 35 clusters.

### 3.3. Prediction of HSCR-Associated miRNAs

According to our workflow (Figure 1), the candidate HSCR genes obtained in this study together with the previously reported HSCR genes obtained from four different sources (Appendix A) were combined in a unique dataset of 691 genes, which was employed to construct the HSCR MTI network from three different experimental validate databases (DIANA, miRecords and miRTarBae), whose individual characteristics are shown in Appendix A.

The number of MRMs, the HSCR-related sub-MRMs and the number of comprised miRNAs and genes present in the sub-MRMs for each MTI network created, after bicluster analysis by BiclusO, are shown in Table 4.

We calculated the total relevance score (TSR) for each miRNA, predicting 137 HSCR-related miRNAs (Appendix A) (with a TSR > 0) out of the initial 430 miRNAs, as shown in Figure 5. As expected, the sum of number of miRNAs encompassed in sub-MRMs obtained from each MTI network (Table 4) was larger than the total number of predicted HSCR-related miRNAs after biclustering and TSR calculation, since a miRNA might be attached to more than one MTI network and because the relevance score (SR) of miRNAs encompassed in a sub-MRM without any direct interaction with a HSCR-related gene is equal to zero.

### 3.4. Assessment of the Relevance of the Identified Candidate miRNAs in HSCR

To validate our results, we searched how many of the predicted miRNAs are exactly matched with well-curated known HSCR miRNAs in five public databases (Appendix A).

Of 137 predicted miRNAs, 17 (12.41%) matched with good-quality known HSCR miRNAs, which, based on a hypergeometric distribution, is significant (*p*-value= 9.01× 10^−8^) if assuming a total number of human mature miRNAs of 2300 [93]. Even if we consider as universal just the total 430 miRNAs present in different MTI datasets before biclustering and TSR calculation, which included 30 HSCR miRNAs, the probability of pulling 17 good-quality known HSCR miRNAs among the list of 137 predicted HSCR-associated miRNAs is also statistically significant (*p*-value = 0.0022). According to literature reports, at least 35 of our predicted miRNAs have been previously described as associated with HSCR onset (Appendix A). Remarkably, among this group of 35 HSCR-related miRNAs, we found 15 of the 17 well-curated known HSCR miRNAs in five public databases (Appendix A). Furthermore, we studied the relevance of the 137 predicted miRNAs in HSCR disease by different aspects.

First, we identified the total number of HSCR genes attached to them in the MTI networks created for this study (Appendix A). Interestingly, it appears that HSCR-related genes are enriched in the top 10 miRNAs. The total number of HSCR genes attached to the top 10 miRNAs was 196, whereas the total number of HSCR genes attached to all 137 miRNAs was 325. Thus, an approximate ratio of 8.27:1 was achieved in terms of attachment to the HSCR genes for the top 10 miRNAs (Figure 6).

Remarkably, we found that five of our predicted miRNAs (hsa-miR-15a-5p, hsa-miR-27b-3p, hsa-miR-195-5p, hsa-miR-218-5p and hsa-miR-128-3p) were RET-regulated miRNAs and two were L1CAM-regulated miRNAs (hsa-miR-1-3p and hsa-miR-34a-5p), but none of them were PHOX2B-regulated miRNA.

Second, we found 3 out of 137 HSCR predicted miRNAs (hsa-miR-200b-3p, hsa-miR-200a-3p and hsa-miR-194-5p) were specifically expressed in the colon.

Third, we found that 9 out of 31 DE miRNAs obtained after DE re-analysis of GSE77296 were present in our list of HSCR associated miRNAs (hsa-miR-142-3p, hsa-miR-148a-3p, hsa-miR-194-5p, hsa-miR-200a-3p, hsa-miR-200b-3p, hsa-miR-200c-3p, hsa-miR-222-3p, hsa-miR-338-3p and hsa-miR-429). Furthermore, we found that another nine of our predicted miRNAs were described as dysregulated in HSCR patients by other citing articles [30,75,76,77] of the GSE77296 dataset (Appendix A), while another ten predicted miRNAs previously associated with HSCR (Appendix A) were reported to have an altered expression in HSCR colon tissue by quantitative techniques [4,5,78,79,80,81,82,83,84]. Altogether, 28 of our predicted miRNAs (20.44%) were described with a dysregulated expression in colon tissue of HSCR patients.

Fourth, we found nine miRNAs previously described as potential peripheral HSCR biomarkers [5,75,81,84,85] that matched with our predicted miRNAs (hsa-miR-181a-5p, hsa-miR-18a-5p, hsa-miR-199a-3p, hsa-miR-200a-3p, hsa-miR-200b-3p, hsa-miR-218-5p, hsa-miR-494-3p, hsa-miR-25-3p and hsa-miR-92a-3p).

Fifth, after having created the miRNA disease network with miRNet, we found that 68 out of the 137 predicted miRNAs were associated with 268 diseases. As expected, we identified HSCR interacting with two miRNAs (hsa-miR-107 and hsa-miR-429) (Appendix A). From the network (Appendix A), the diseases with highest node connectivity degree (≥10) were related both to different kinds of cancer (e.g., hepatocellular carcinoma, lung cancer, prostate cancer and colorectal cancer), and other diseases, such as cardiac hypertrophy and Duchenne muscular dystrophy (both related to muscular dystrophies), Alzheimer disease (a neurodegenerative disease associated with colonic dysmotility/inflammation in its earliest stages) and ulcerative colitis (a chronic inflammatory bowel disease).

Sixth, we found that six of our predicted miRNAs (hsa-miR-140-3p; hsa-miR-142-3p; hsa-miR-324-5p; hsa-miR-326; hsa-miR-338-3p; hsa-miR-944) were predicted targets of dysregulated circRNAs in HSCR (Appendix A).

Seventh, we found 27 HSCR-related lncRNA-miRNA interactions involving 27 of the miRNAs predicted as HSCR-associated in this study after constructing a miRNA-HSCR-related lncRNA network in miRNet (Appendix A). Interestingly, MEG3 and other dysregulated circRNAs identified in HSCR were predicted to act as sponges of hsa-miR-326.

Finally, over-representation analysis (ORA) was performed on our miRNAs identified by miEAA 2.0. Significant enrichment terms from twenty-six categories were obtained. The main results of interest are shown in Appendix A, to know the following: For the category Chromosomal location (miRBase), we found three over-represented enrichment terms (Chromosome 13, Chromosome 9 and Chromosome 1) that were statistically significant (adjusted *p*-value < 0.05). In the category exRNA forms (miRandola), none of our predicted miRNAs were complex with high-density lipoprotein (HDL), and only 14 of our predicted miRNAs were not significantly enriched in any other extracellular form (hsa-miR-1826, hsa-miR-199b-3p, hsa-miR-211-5p, hsa-miR-296-3p, hsa-miR-320a, hsa-miR-448, hsa-miR-449c-5p, hsa-miR-506-3p, hsa-miR-509-3-5p, hsa-miR-548h-3p, hsa-miR-548j-5p, hsa-miR-548z, hsa-miR-675-3p and hsa-miR-944). For the category Cell-type specific (Atlas), the only significant enrichment term was Neuronal stem cell, with five observed miRNAs associated (hsa-miR-107, hsa-miR-124-3p, hsa-miR-128-3p, hsa-miR-181d-5p and hsa-miR-7-5p) and an adjusted *p*-value = 0.04. For the category Expressed in tissue (Tissue Atlas), we found colon in the top three most over-represented significant terms, with 114 of our predicted miRNAs associated. Considering that HSCR is a born defect often diagnosed in the newborn period and with a significant sex bias, we carefully evaluated the result obtained for the category Gender and Age. Interestingly, we found statistically significant over-representation of the terms “negatively correlated with age” and “upregulated in males”, while the terms “positively correlated with age” and “upregulated in females” were under-represented (Appendix A), which was consistent with the sex bias observed in HSCR patients. As expected, in the category Diseases (MNDR), we found Hirschsprung’s disease (adjusted *p*-value = 0.005) with five observed miRNAs (hsa-miR-107, hsa-miR-142-3p, hsa-miR-195-5p, hsa-miR-338-3p and hsa-miR-429), with the miRNAs previously identified in our disease-miRNA network constructed by miRNet also included.

On the other hand, 65 HSCR-associated pathways from CTD were enriched terms in our predicted miRNAs (Table 5). Interestingly, 11 out of these HSCR-associated pathways (Table 5) were in the Top 20 of the most significant enriched terms in the categories KEGG (miRPathDB) and Reactome (miRPathDB), to know cellular responses to stress, cellular senescence, cytokine signaling in immune system, disease, diseases of signal transduction, immune system, microRNAs in cancer, pathways in cancer, PIP3 activates AKT signaling, signal transduction and signaling by interleukins. Moreover, we looked for the prolactin signaling pathway (with 37 observed miRNAs and an adjusted *p*-value = 4.68 × 10^−13^), and aldosterone-regulated sodium reabsorption (with nine observed miRNAs and an adjusted *p*-value = 0.002) among the statistically significant pathways for the category KEGG (miRPathDB), confirming that these proposed HSCR-related pathways were also enriched in the predicted miRNAs.

Additionally, Table 6 shows the enriched terms significantly over-represented obtained for the category Annotation (gene ontology). Remarkably, 14 of these significant enriched GO terms were associated with HSCR.

### 3.5. Potential miRNA Biomarker Identification in HSCR Based on miRNA-Target Regulatory Network Analyses

We generated a reference HSCR-specific MTI network based on DE miRNAs/genes in HSCR and HSCR-known associated miRNAs/genes from five different databases [32,33,34,35,36,37]. The reference HSCR-specific MTI network contained 129,139 microRNA-target regulatory pairs. After calculating NSR and TFP parameters, miRNAs with significantly high values were highlighted (Table 7).

Overall, 12 predicted miRNAs in this study were screened as candidate biomarkers. Except hsa-miR-30a-5p, all of them have been previously described with a dysregulated expression in HSCR colon tissue. Among them, three were present as DE miRNAs in GSE77296 (hsa-miR-200b-3p, hsa-miR-148a-3p and hsa-miR-429). However, when we used as input the list of DE miRNAs in GSE77296 to screen their potential as biomarkers, we only found two candidates (hsa-miR-146a-5p and hsa-miR-200b-3p). Therefore, the only overlapped miRNA was hsa-miR-200b-3p.

## 4. Discussion

Network-based computational approaches are being used to identify disease-associated subnetworks and for efficient disease–gene association prediction [94]. Biological network studies are based on two underlying assumptions: Firstly, human diseases are the consequences of disruption in molecular networks [95], and secondly, genes associated with the same or similar diseases tend to reside in the same neighborhood of a PPI network [96,97] or a pathway [98].

Network-based methods have been successfully applied in the study of rare diseases [99,100,101,102], and specific methodology has been developed for searching and ranking disease-causing genes in this kind of disorders [103,104,105]. According to Orphanet information, 71.9% of rare diseases are genetic and 69.9% are exclusively pediatric onset [106], such as HSCR. However, HSCR and other rare conditions are characterized by a wide diversity of symptoms and signs that vary from patient to patient, and their pathogenesis is not fully understood because of our partial knowledge of the molecular basis and causative genes [4]. For this reason, in this study, we aimed to test if reported methods [49,57] previously validated in the context of inflammatory bowel disease (IBD, a non-rare disease) can be generalized to find disease modules and disease-associated genes for HSCR.

For the employed of the first method [49], a condition-specific PPI network was constructed by mapping the gene expression data obtained in transcriptome and proteome experiments onto the global PPI network, and by locating well-known disease genes (deposited in DisGeNET [33], which collects disease–gene association obtained from experimental assays such as GWAS, NGS, etc.) in the network. This strategy allowed us to propose up to 178 HSCR-related genes and validate this approach after identifying 14 well-curated HSCR genes among our candidates. Therefore, these results could involve an increase in the number of genes potentially involved in HSCR of up to 33.78% considering the 527 HSCR genes according to the four used databases, allowing us to guide and focus subsequent studies. This value is considerably higher than that obtained by other strategies also aiming to identify new HSCR-associated genes [14,15,16,17], which shows the efficacy of this cost-free study, even without being exempt from certain limitations and weaknesses, such as (i) the lack of consideration of expression differences between the postnatal stenotic tissue (our input) and the developing fetus in which the neurogenesis occurs [28]; (ii) the limited number of published high-throughput omics expression studies in HSCR and the absence of these types of data in public repositories [107,108,109,110]; (iii) the absence of demographic information and clinical disease parameters from all participants of the included studies [28,29,30,31]; and (iv) the employment of only some of the available integrative database resources for disease–gene associations [32,33,34,35,36,111] both for gene–disease module identification and result validation.

This enrichment analysis revealed that a high percentage of our candidate genes were involved in previously HSCR-associated pathways and GO terms [16,17]. In this line, it is noteworthy that approximately 30% of the 178 candidate genes were related to cell adhesion processes (Table 3). This biological functionality is related to HSCR across different patient populations [16]. In a biological context, cell adhesion molecules (CAMs) and fibroblast growth factors (FGFs) are known to stimulate neurite growth through the activation of various FGF receptors (FGFRs) on neurons [112]. The marked decrease in CAM expression in the stenotic intestine of HSCR patients suggests that this alteration would lead to defects in the migration of enteric neuroblasts [112]. Furthermore, approximately 20% of these candidate genes were described as tumor suppressor and/or oncogenes in any context (Appendix A), which is interesting considering that all clinical gene panels for HSCR in ClinGen have been suggested as important players in the development of human carcinomas [113,114,115,116]. In this line, HSCR has been reported in the co-occurrence of several kinds of cancers, such as familial medullary thyroid carcinoma, multiple endocrine neoplasia type 2A and type 2B [117], neuroblastoma [118,119] hepatoblastoma [120,121], central nervous system tumors [122] and retinoblastoma [123,124].

Moreover, this approach also allowed us to identify 55 significant disease gene modules enriched in known HSCR DE genes in patient colon tissue samples. As expected, comparative enrichment analysis showed that several of the significantly enriched path-ways present in more than 50% of these clusters (Figure 4) were related to previously HSCR-associated pathways, such as VEGF signaling, which results in the activation of multiple downstream pathways, including the Ras/MAPK pathways [125,126] (pathways systematically associated with HSCR [16,17]), regulating cell proliferation and gene expression; and/or the PLCγ pathway, controlling vascular permeability [127]. In this line, despite there being no reports of increased vascular permeability in the gastrointestinal tract (GIT) of HSCR patients, a recent study has suggested that vascular permeability may be etiologic for postoperative Hirschsprung-associated enterocolitis (HAEC) [128]. In addition, vascular endothelial growth factor A (VEGF-A) mediates angiogenesis, and altered vascular density has also been reported in GIT of HSCR patients and in some mouse models of HSCR [129]. However, contradictory results were observed when comparing mouse and human data, which might be due to several reasons, to know that (a) the human data were not standardizable; (b) for the analysis of the vascular density in colon samples of HSCR patients, only one area per sample (which exhibited the highest numbers of microvessels) was investigated; and (c) the statistical analysis was performed using a Mann–Whitney test on two dependent samples because analyzed ganglionic and aganglionic colon tissue samples belong for the same group of HSCR patients. Therefore, in spite of the relevance of these preliminary results, their specific clinical importance in HSCR is as yet unclear [128].

Additionally, other two KEGG pathways—the prolactin signaling pathway and aldosterone-regulated sodium reabsorption pathway—were proposed as candidate HSCR-related pathways. Prolactin-like material (PLM) has been previously identified in the gut mucosa, but is absent for aganglionic segment of HSCR patients [130]. In spite of the lack of experimental validation assays, it has been suggested that, as pituitary prolactin, PLM might be involved in metabolism and osmoregulation [130], like the aldosterone-regulated sodium reabsorption pathway (Figure 4b). This pathway is an electrogenic sodium transport through the amiloride-sensitive epithelial sodium channel (*ENaC*) [131]. Along the intestine, electrogenic absorption through EnaC is limited to surface epithelial cells of the distal colon and rectum [132]. The distal aganglionic colon segment in HSCR patients leads to abdominal distention with intraluminal loss of chloride, potassium, and sodium, resulting in metabolic alkalosis with secondary hyperreninemia and hyperaldosteronism [133]. However, after proctocolectomy, *EnaC* starts to be expressed in the distal part of the small intestine. Therefore, possible symptoms of the pseudo-Bartter syndrome in neonates with HSCR completely disappeared after a pull-through operation with the removal of the causal factor [133]. The exact importance of this pathway in HSCR has not been clarified yet but it is known that Endothelin receptor type B (*EDNRB*), the second most mutated gene in Hirschsprung (5% cases) [1], is an antagonist of *EnaC* activity [134]. Moreover, the downregulation of *EnaC* with a reduction in sodium reabsorption in the colon may contribute to diarrhea associated with IBD [131], a disease with relatively low co-occurrence in HSCR patients with a similar clinical presentation to HAEC [135,136].

Recently, miRNAs have received widespread attention due to their role in the regulation of cellular processes [137] and the increase in experimental evidence postulating that their aberration can be among the triggering causes of pathological phenomena, including HSCR [85,138,139,140]. Additionally, miRNAs are endogenous, stable, easily accessible in the extracellular space and detectable in patients’ tissue and blood specimens [85]. Therefore, several authors have suggested that they may have utility as diagnostic and prognostic biomarkers for HSCR [5]. For these reasons, our second aim was to decipher key miRNAs related to HSCR by using computational tools to unravel miRNA–disease associations and to identify miRNA biomarkers.

The application of biclustering algorithms can solve the traditional problems associated with the discovery of regulatory modules that control gene transcription in biological model [141]. The biclustering algorithm of BiClusO [54,59,60], as well as the one employed in ComiRNet [142], allowed for the efficient discovery of overlapping and highly cohesive biclusters. Then, an overall relevance score (TSR) [57] based on the exploration of only those biclusters containing genes of interest (known disease-associated genes) was applied to predict Hirschsprung-related miRNAs in this study. The use of this second previously reported method, validated for IBD, is suitable for isolating miRNAs for similar types of diseases [57]. Although HSCR was not identified in its miRNA–disease network, as we mentioned above, the co-occurrence of IBD, especially Crohn’s disease, has been recently reported in children with HSCR [133,134], and common pathogenesis and hub gene biomarkers have been also identified [143].

In fact, the application of this method allowed us to predict 137 HSCR-related miRNAs and validated the method, which is especially relevant for a rare disease with a limited knowledge of its genetic landscape. Remarkably, we found five RET-regulated miRNAs (hsa-miR-15a-5p, hsa-miR-27b-3p, hsa-miR-195-5p, hsa-miR-218-5p and hsa-miR-128-3p) and two L1CAM-regulated miRNAs (hsa-miR-1-3p and hsa-miR-34a-5p) among our predicted HSCR-related miRNAs. Moreover, we identified three miRNAs with specific colon tissue expression among our predicted miRNAs, with has-miR-194-5p being the only one without a detailed study of its implication in HSCR to our knowledge [75].

However, when we tried to evaluate the relevance of the predicted miRNAs in the disease context, we noticed some methodological considerations that should be mentioned: Firstly, a majority of microRNA–mRNA interactions remain unidentified [144]. However, to avoid a high rate of false positives among our predicted miRNAs, we refused the use of predictive algorithms, and only considered experimentally validated miRNA-target interaction [58]. Secondly, other molecules such as lncRNAs and circRNAs can affect the functions of miRNAs in gene expression and play an important role in regulating cell physiological functions and diseases [145]. Therefore, an integrative analysis of the lncRNA/circRNA-miRNA-mRNA ceRNA network would be required. However, the current availability of lncRNA and circRNA expression data in HSCR patients is not enough to carry it out, although our results also pointed to the interconnection of these molecules in regulatory modules. Concretely, we found that six of our predicted miRNAs were predicted targets of dysregulated circRNA in HSCR. Nevertheless, only hsa-miR-142-3p and hsa-miR-338-3p have been described as significantly upregulated in stenotic segment tissues of HSCR patients [30]. Of note, several experimental validated target mRNAs of these two miRNAs were also dysregulated in intestine samples of HSCR patients and were related to the pathophysiology of this disease, such as *ATP2A2*, *CSE1L*, *LRP8* and *THBS4* (target of hsa-miR-142-3p) and *CDH2* and *PKM* (target of hsa-miR-338-3p). Finally, the software tool employed for miRNA biomarker identification is based on just two significant parameters, but it lacks a disease-specific signal in this approach [61]. Nonetheless, all putative biomarkers obtained (Table 7) from our list of predicted HSCR-associated miRNAs were involved in disease pathogenesis according to well-curated known HSCR miRNAs in five public databases and literature reports [30] (Appendix A).

On the other hand, functional analysis of predicted miRNAs revealed the meaningful results obtained in this study. These results will serve as a valuable resource for further exploration by experiments. For example, it is potentially important that the putative biomarkers originally identified were able to be validated in blood samples, since blood could be easily obtained for clinical translation [5,84].

## 5. Conclusions

In summary, large-scale high-dimensional omics expression data analyses have been successfully applied to the discovery of functional connections among diseases, genetic perturbation, and drug action. However differential expression analysis alone does not provide any functional insights. For this reason, we designed the bioinformatics framework proposed in this study, which has allowed us to decipher putative modules, genes and miRNAs in HSCR, which could open new lines of research on this disease. In this line, this study has identified new candidate genes and miRNAs not previously associated with HSCR, but which are involved in important biological processes related to enteric nervous system development or HSCR onset. The employed approach in this study also was useful for amplification and facilitating research into HSCR specific-functional connections such as the proposed HSCR pathways: the prolactin signaling pathway and aldosterone-regulated sodium reabsorption pathway. Therefore, we consider that this methodology can be also successfully applied to other rare complex diseases for which conducting large, randomized trials is difficult.

## Figures and Tables

**Figure 1 biomolecules-14-00164-f001:**
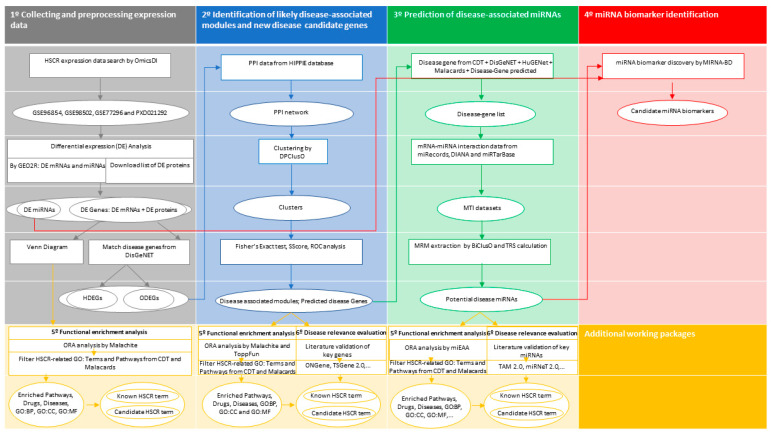
Flowchart containing the main steps followed on this study. The diagram is divided into four major sequential blocks, listed one through four. Additionally, the complementary working packages “5° Functional enrichment analysis” and “6° Disease relevance evaluation” are shown in yellow. Arrows show the direction of the workflow. Processes are represented by a rectangle, while the ovals correspond to the input/output data of each process. HSCR, Hirschsprung disease; OmicsDI, Omics Discovery Index; GEO2R, interactive web tool of Gene Expression Omnibus (GEO); miRNA, microRNA; HDEGs, known HSCR DE genes; ODEGs, other DE genes; ORA, over-representation analysis; CDT, Comparative Toxicogenomics Database; BP, biological process; MF, molecular function; CC, cellular component; PPI, protein–protein interaction; HIPPIE, Human Integrated Protein–Protein Interaction reference database; SScore, significance score; ROC, receiver operating characteristic; MTI, MiRNA–target interaction; MRM, miRNA-regulatory module; TSR, total relevance score; miEAA, the miRNA Enrichment Analysis and Annotation Tool.

**Figure 2 biomolecules-14-00164-f002:**
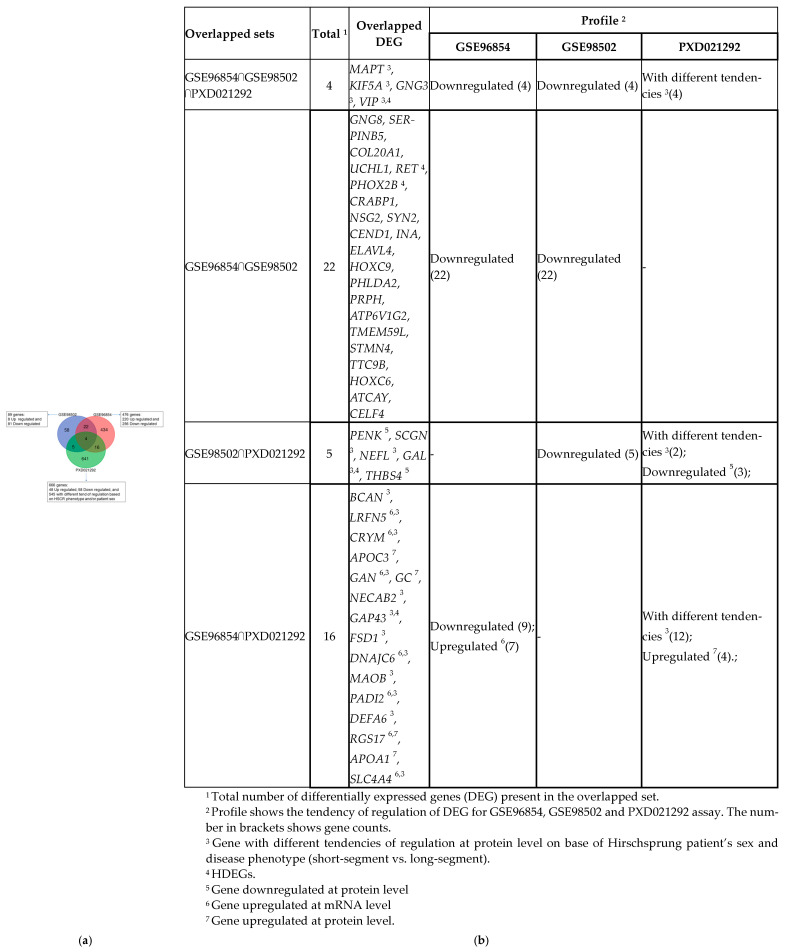
Gene overlapping bioinformatic analysis. (**a**) Venn diagram showing the number of common differentially expressed genes among the ganglionic and stenotic colon segment groups from the transcriptomic (GSE96854 and GSE98502) and proteomic (PXD021292) experiments; (**b**) Table depicting the detailed information of the overlap genes that were significantly regulated in more than one sequencing experiment.

**Figure 3 biomolecules-14-00164-f003:**
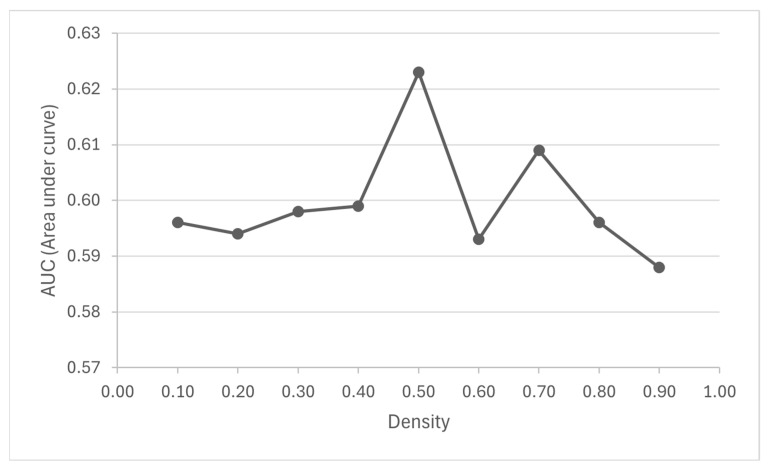
Plot of AUCs corresponding to nine density sets of clusters after ROC analysis.

**Figure 4 biomolecules-14-00164-f004:**
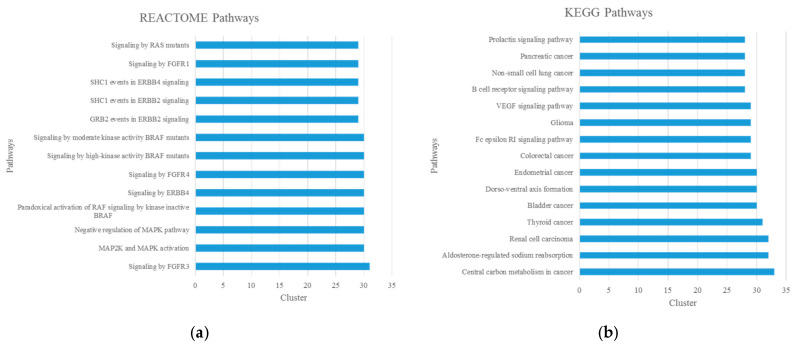
Enrichment pathways analysis of significant gene clusters obtained with Malachite in this study. (Cluster ≥ 28, for 28 pathways): (**a**) REACTOME pathways; (**b**) KEGG pathways.

**Figure 5 biomolecules-14-00164-f005:**
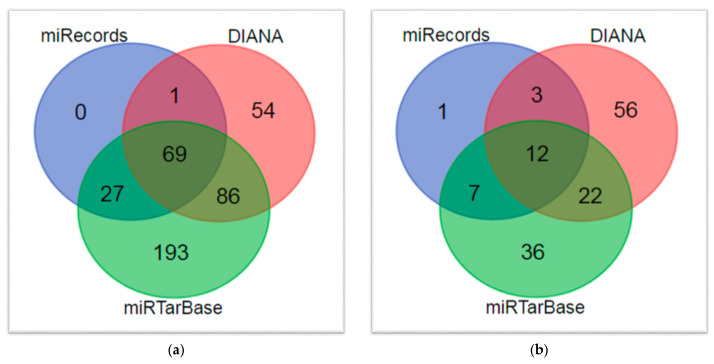
Total miRNAs in different MTI datasets. (**a**) Before biclustering; (**b**) predicted after biclustering and TSR calculation.

**Figure 6 biomolecules-14-00164-f006:**
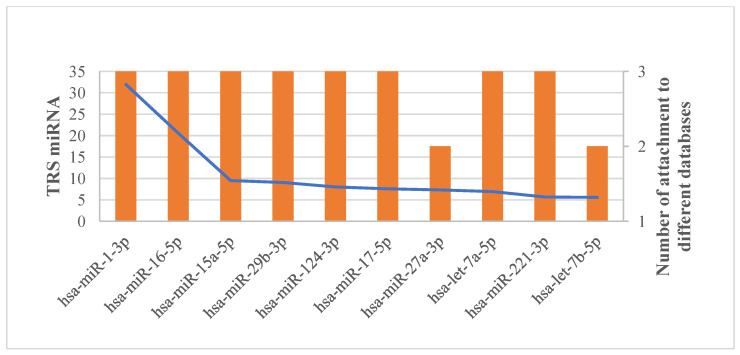
Total score (TRS) of top 10 predicted HSCR-related miRNAs with the number of attachments to different databases.

**Table 1 biomolecules-14-00164-t001:** Topological parameters of the protein–protein interaction (PPI) network constructed in the present study obtained with NAP.

Characteristic	PPI ^1^ Network
Number of edges ^2^	14,343
Number of nodes ^3^	3806
Diameter ^4^	11.00
Average path length ^5^	3.83
Clustering coefficient ^6^	0.05
Modularity ^7^	0.42
Number of self loops ^8^	269.00
Average eccentricity ^9^	8.16
Average eigenvector centrality ^10^	0.03
Average number of neighbors ^11^	0.97
Centralization betweenness ^12^	0.18
Centralization degree ^13^	0.14
Fit power law ^14^	TRUE

^1^ PPI, Protein–protein interaction. ^2^ Number of edges: shows the number of connections in the PPI network. ^3^ Number of nodes: shows the number of vertices in the PPI network. ^4^ Diameter: shows the length of the longest geodesic. ^5^ Average path length: the average number of steps needed to go from a node to any other. ^6^ Clustering coefficient: a metric to show if the network has the tendency to form clusters. ^7^ Modularity: this function calculates how modular is a given division of a graph into subgraphs. ^8^ Number of self-loops: how many nodes are connected to themselves. ^9^ Average eccentricity: the eccentricity of a vertex is its shortest path distance from the farthest other node in the graph. ^10^ Average eigenvector centrality: this shows the influence of a node in a network. ^11^ Average number of neighbors: how many neighbors each node of the network has on average. ^12^ Centralization betweenness: betweenness centrality quantifies the number of times a node acts as a bridge along the shortest path between two other nodes. ^13^ Centralization degree: This is defined as the number of links incident upon a node. ^14^ Fit power law: the degree distribution of proteins approximates a power law 𝒫(k) = k^−γ^.

**Table 2 biomolecules-14-00164-t002:** Results of each DPClusO clustering based on the HSCR relevant PPI network.

Density	Total Cluster	Max Size	Avg. Size	Significant Cluster Count
0.1	404	148	25.74	59
0.2	770	73	12.67	61
0.3	1213	47	8.2	257
0.4	1639	37	6.2	464
0.5	2093	27	4.94	563
0.6	3602	19	2.46	962
0.7	3796	17	2.49	1051
0.8	3851	12	2.39	1056
0.9	3861	8	2.38	1070

**Table 3 biomolecules-14-00164-t003:** Top 10 significant pathways and GO terms obtained after enrichment analysis of predicted Hirschsprung-related genes with ToppFun.

Enriched Category	ID	Name	FDR ^1^	Gene Count
GO: Molecular Function	GO:0050839	Cell adhesion molecule binding	6.15 × 10^−21^	43
	GO:0045296	Cadherin binding	4.14 × 10^−19^	33
	GO:0019901	Protein kinase binding	2.34 × 10^−15^	43
	GO:0019900	Kinase binding	1.50 × 10^−14^	44
	GO:0044877	Protein-containing complex binding	1.27 × 10^−11^	54
	GO:0019904	Protein domain specific binding	7.72 × 10^−5^	30
	GO:0005102	Signaling receptor binding ^2^	2.25 × 10^−4^	43
	GO:0005158	Insulin receptor binding	7.29 × 10^−4^	7
	GO:1990782	Protein tyrosine kinase binding	9.09 × 10^−4^	12
	GO:0030527	Structural constituent of chromatin	2.01 × 10^−3^	10
GO: Biological Process	GO:0051094	Positive regulation of developmental process	2.14 × 10^−12^	57
	GO:0002009	Morphogenesis of an epithelium	4.09 × 10^−12^	42
	GO:0000902	Cell morphogenesis ^2^	1.05 × 10^−10^	48
	GO:0048729	Tissue morphogenesis	1.05 × 10^−10^	43
	GO:0007155	Cell adhesion ^2^	1.25 × 10^−10^	53
	GO:0022603	Regulation of anatomical structure morphogenesis	2.32 × 10^−10^	45
	GO:0040011	Locomotion	7.88 × 10^−10^	53
	GO:0043067	Regulation of programmed cell death	1.61 × 10^−9^	54
	GO:0030155	Regulation of cell adhesion ^2^	1.61 × 10^−9^	38
	GO:0042981	Regulation of apoptotic process	2.35 × 10^−9^	53
GO: Cellular Component	GO:0031252	Cell leading edge	2.28 × 10^−15^	34
	GO:0070161	Anchoring junction	1.65 × 10^−14^	52
	GO:0098590	Plasma membrane region ^2^	1.18 × 10^−13^	52
	GO:0045121	Membrane raft ^2^	2.43 × 10^−13^	30
	GO:0098857	Membrane microdomain	2.43 × 10^−13^	30
	GO:0030027	Lamellipodium	3.46 × 10^−13^	23
	GO:0005911	Cell–cell junction	5.31 × 10^−13^	33
	GO:0043005	Neuron projection	1.67 × 10^−11^	53
	GO:0030055	Cell-substrate junction	6.67 × 10^−11^	27
	GO:0005925	Focal adhesion	2.74 × 10^−10^	26
KEGG Pathway	83070	Adherens junction	4.01 × 10^−9^	15
	782000	Proteoglycans in cancer	8.18 × 10^−7^	19
	585563	Alcoholism	4.68 × 10^−5^	16
	83083	Leukocyte transendothelial migration	7.43 × 10^−5^	13
	868086	Rap1 signaling pathway	2.66 × 10^−4^	16
	83109	Endometrial cancer	3.84 × 10^−4^	9
	83067	Focal adhesion	6.72 × 10^−4^	15
	101143	Neurotrophin signaling pathway	7.67 × 10^−4^	12
	946598	Thyroid hormone signaling pathway	3.57 × 10^−3^	11
	117293	Arrhythmogenic right ventricular cardiomyopathy (ARVC)	4.35 × 10^−3^	9
Reactome Pathway	1269326	Interleukin-7 signaling	2.00 × 10^−7^	10
	1269507	Signaling by Rho GTPases	3.61 × 10^−7^	27
	1269512	RHO GTPases activate PKNs	1.06 × 10^−6^	14
	1269509	RHO GTPase effectors	1.16 × 10^−6^	22
	1270302	Developmental biology ^2^	2.33 × 10^−6^	41
	1269340	Hemostasis	1.14 × 10^−5^	30
	1269811	Mitotic prophase	1.65 × 10^−5^	15
	1270437	HDMs demethylate histones	4.72 × 10^−5^	10
	1269318	Signaling by interleukins ^2^	6.46 × 10^−5^	26
	1269602	Formation of the beta-catenin:TCF transactivating complex	7.43 × 10^−5^	12
PID Pathway	138071	PDGFR-beta signaling pathway	6.46 × 10^−5^	10
	137930	Signaling events mediated by hepatocyte growth factor receptor (c-Met)	7.20 × 10^−5^	11
	137940	Signaling events mediated by VEGFR1 and VEGFR2	2.35 × 10^−4^	10
	137919	N-cadherin signaling events	2.58 × 10^−4^	8
	137970	EGF receptor (ErbB1) signaling pathway	6.17 × 10^−4^	7
	169348	Signaling events mediated by focal adhesion kinase	6.52 × 10^−4^	9
	137989	FGF signaling pathway	2.31 × 10^−3^	8
	138017	Signaling events mediated by PTP1B	3.37 × 10^−3^	8
	137915	Signaling events regulated by Ret tyrosine kinase ^2^	6.52 × 10^−3^	7
	137977	Neurotrophic factor-mediated Trk receptor signaling	6.83 × 10^−3^	8

^1^ FDR, false discovery rate. ^2^ Hirschsprung-related enriched term.

**Table 4 biomolecules-14-00164-t004:** Summary of BiclusO clustering results obtained in this study for each Hirschsprung MTI network.

MTI Network ^1^	MRM ^2^	Sub-MRM ^3^	miRNA ^4^	Gene ^5^
DIANA	636	266	98	1160
miRTarbase	58	23	81	407
miRecords	16	13	24	143

^1^ MTI, miRNA–target interaction. ^2^ MRM, miRNA-regulatory module. ^3^ sub-MRM, Hirschsprung related bicluster. ^4^ miRNA, microRNA comprised in sub-MRMs. ^5^ Gene, number of genes comprised in sub-MRMs.

**Table 5 biomolecules-14-00164-t005:** Significant enriched HSCR-associated pathways of predicted HSCR-related miRNAs.

Category	Subcategory	Pathway ID	Adjusted *p*-Value	Count ^1^
Reactome (miRPathDB)	Cellular senescence	R-HSA-2559583	3.73 × 10^−34^	71
	Signal transduction	R-HSA-162582	5.54 × 10^−33^	66
	Cellular responses to stress	R-HSA-2262752	1.95 × 10^−30^	71
	PIP3 activates AKT signaling	R-HSA-1257604	5.59 × 10^−26^	57
	Cytokine signaling in immune system	R-HSA-1280215	1.30 × 10^−24^	53
	Signaling by interleukins	R-HSA-449147	3.74 × 10^−22^	51
	Disease	R-HSA-1643685	3.79 × 10^−21^	45
	Diseases of signal transduction	R-HSA-5663202	2.37 × 10^−20^	46
	Immune system	R-HSA-168256	5.19 × 10^−20^	45
	Oxidative stress-induced senescence	R-HSA-2559580	1.73 × 10^−18^	51
	Developmental biology	R-HSA-1266738	5.94 × 10^−16^	41
	Signaling by PTK6	R-HSA-8848021	3.90 × 10^−14^	32
	Post-translational protein modification	R-HSA-597592	1.37 × 10^−12^	34
	Signaling by ERBB2	R-HSA-1227986	2.24 × 10^−11^	26
	Metabolism of proteins	R-HSA-392499	2.49 × 10^−11^	31
	VEGFA-VEGFR2 pathway	R-HSA-4420097	1.15 × 10^−9^	20
	SUMOylation	R-HSA-2990846	2.08 × 10^−9^	24
	Signaling by EGFR	R-HSA-177929	7.80 × 10^−9^	15
	SUMO E3 ligases SUMOylate target proteins	R-HSA-3108232	8.52 × 10^−9^	23
	Downregulation of ERBB2 signaling	R-HSA-8863795	4.76 × 10^−8^	18
	Adaptive immune system	R-HSA-1280218	6.79 × 10^−8^	19
	Signaling by VEGF	R-HSA-194138	6.79 × 10^−8^	19
	MAPK family signaling cascades	R-HSA-5683057	8.15 × 10^−8^	23
	SHC1 events in ERBB2 signaling	R-HSA-1250196	3.96 × 10^−7^	15
	Axon guidance	R-HSA-422475	1.11 × 10^−6^	22
	SHC1 events in EGFR signaling	R-HSA-180336	8.22 × 10^−6^	12
	SHC1 events in ERBB4 signaling	R-HSA-1250347	1.86 × 10^−5^	10
	RET signaling	R-HSA-8853659	3.17 × 10^−5^	8
	GRB2 events in ERBB2 signaling	R-HSA-1963640	5.86 × 10^−5^	10
	Signaling to RAS	R-HSA-167044	6.64 × 10^−5^	8
	Signaling by ERBB4	R-HSA-1236394	8.88 × 10^−5^	13
	Downstream signal transduction	R-HSA-186763	1.13 × 10^−4^	12
	SOS-mediated signaling	R-HSA-112412	1.53 × 10^−4^	9
	Innate immune system	R-HSA-168249	1.64 × 10^−4^	13
	ERBB2 activates PTK6 signaling	R-HSA-8847993	3.34 × 10^−4^	7
	ERBB2 regulates cell motility	R-HSA-6785631	3.83 × 10^−4^	9
	GRB2 events in EGFR signaling	R-HSA-179812	3.83 × 10^−4^	9
	PI3K events in ERBB2 signaling	R-HSA-1963642	6.03 × 10^−4^	7
	PI3K events in ERBB4 signaling	R-HSA-1250342	6.03 × 10^−4^	4
	GRB7 events in ERBB2 signaling	R-HSA-1306955	9.79 × 10^−4^	5
	Signaling by SCF-KIT	R-HSA-1433557	1.26 × 10^−3^	10
	Nuclear signaling by ERBB4	R-HSA-1251985	1.57 × 10^−3^	6
	Constitutive signaling by aberrant PI3K in cancer	R-HSA-2219530	1.65 × 10^−3^	10
	rRNA processing	R-HSA-72312	1.65 × 10^−3^	10
	FCERI mediated MAPK activation	R-HSA-2871796	1.83 × 10^−3^	12
	Gastrin-CREB signaling pathway via PKC and MAPK	R-HSA-881907	2.12 × 10^−3^	9
	Signaling by PDGF	R-HSA-186797	6.39 × 10^−3^	7
	Signaling to ERKs	R-HSA-187687	8.02 × 10^−3^	10
	IGF1R signaling cascade	R-HSA-2428924	1.86 × 10^−2^	8
	IRS-related events triggered by IGF1R	R-HSA-2428928	1.86 × 10^−2^	8
	VEGFR2 mediated cell proliferation	R-HSA-5218921	2.22 × 10^−2^	8
	Signaling by leptin	R-HSA-2586552	2.25 × 10^−2^	3
	Interleukin receptor SHC signaling	R-HSA-912526	2.43 × 10^−2^	2
	DAP12 signaling	R-HSA-2424491	3.44 × 10^−2^	6
	rRNA processing in the nucleus and cytosol	R-HSA-8868773	3.50 × 10^−2^	7
	SUMOylation of DNA damage response and repair proteins	R-HSA-3108214	4.08 × 10^−2^	6
	SUMOylation of RNA-binding proteins	R-HSA-4570464	4.08 × 10^−2^	7
	Signaling by insulin receptor	R-HSA-74752	4.08 × 10^−2^	7
KEGG (miRPathDB)	MicroRNAs in cancer	hsa05206	1.68 × 10^−42^	98
	Pathways in cancer	hsa05200	6.85 × 10^−28^	78
	ErbB signaling pathway	hsa04012	4.68 × 10^−13^	40
	Thyroid cancer	hsa05216	1.29 × 10^−11^	33
	Transcriptional misregulation in cancer	hsa05202	1.47 × 10^−7^	34
	Endocytosis	hsa04144	4.77 × 10^−6^	22
	Melanogenesis	hsa04916	3.44 × 10^−4^	12

^1^ Total number of observed miRNAs.

**Table 6 biomolecules-14-00164-t006:** Enriched terms significantly over-represented obtained for the category Annotation (gene ontology) of predicted HSCR-associated miRNAs with miEAA.

Subcategory	Adjusted *p*-Value	Count ^1^
Gene silencing by miRNA GO:0035195	1.19 × 10^−10^	98
mRNA binding involved in post-transcriptional gene silencing GO:1903231	6.99 × 10^−9^	97
MiRNA mediated inhibition of translation GO:0035278	5.86 × 10^−6^	39
Negative regulation of cell population proliferation GO:0008285 ^2^	7.40 × 10^−4^	17
Negative regulation of gene expression GO:0010629 ^2^	7.40 × 10^−4^	21
Positive regulation of ERK1 and ERK2 cascade GO:0070374 ^2^	7.40 × 10^−4^	9
Extracellular space GO:00056152	8.34 × 10^−4^	84
Positive regulation of connective tissue replacement GO:1905205 ^2^	1.99 × 10^−3^	8
Extracellular exosome GO:0070062 ^2^	3.66 × 10^−3^	15
Negative regulation of cardiac muscle cell apoptotic process GO:0010667 ^2^	4.25 × 10^−3^	10
Positive regulation of vascular smooth muscle cell proliferation GO:1904707	4.25 × 10^−3^	12
Negative regulation of apoptotic process GO:0043066 ^2^	4.96 × 10^−3^	7
Negative regulation of angiogenesis GO:0016525	7.43 × 10^−3^	19
Positive regulation of angiogenesis GO:0045766	7.47 × 10^−3^	13
Negative regulation of sprouting angiogenesis GO:1903671	8.07 × 10^−3^	12
Negative regulation of cell migration GO:0030336 ^2^	9.15 × 10^−3^	15
Nucleus GO:0005634 ^2^	1.30 × 10^−2^	6
Positive regulation of apoptotic process GO:0043065 ^2^	1.82 × 10^−2^	11
Cytoplasm GO:0005737 ^2^	2.09 × 10^−2^	8
Positive regulation of protein kinase B signaling GO:0051897 ^2^	2.09 × 10^−2^	8
Cellular response to vascular endothelial growth factor stimulus GO:0035924	3.53 × 10^−2^	5
Negative regulation of I-kappaB kinase/NF-kappaB signaling GO:0043124	3.53 × 10^−2^	5
Positive regulation of metalloendopeptidase activity GO:1904685	3.53 × 10^−2^	5
Negative regulation of protein kinase B signaling GO:0051898	3.60 × 10^−2^	10
Negative regulation of cholesterol efflux GO:0090370 ^2^	4.11 × 10^−2^	9
Negative regulation of low-density lipoprotein particle clearance GO:0010989	4.40 × 10^−2^	6
Positive regulation of cardiac muscle hypertrophy in response to stress GO:1903244	4.40 × 10^−2^	6
Negative regulation of G1S transition of mitotic cell cycle GO:2000134	4.85 × 10^−2^	11
Negative regulation of inflammatory response GO:0050728	4.85 × 10^−2^	15

^1^ Total number of miRNAs observed. ^2^ HSCR-related GO term.

**Table 7 biomolecules-14-00164-t007:** Prediction values for candidate miRNA biomarkers identified for Hirschsprung disease (HSCR) based on the model with the reconstructed HSCR miRNA-mRNA reference network. Values of single-line regulation (NSR) and transcription factor percentage (TFP) parameters were calculated to characterize the regulatory pattern of the miRNAs in the constructed reference network and to identify miRNAs that could have a crucial role in the progression of HSCR for each (a) HSCR miRNA predicted in this study; (b) differentially expressed (DE) miRNA in GSE77296.

Predicted HSCR-Related miRNAs in This Study	miRNA Symbol	NSR Value	*p*-Value of NSR	TFP Value	*p*-Value of TFP
	hsa-let-7a-5p	113	1.88 × 10^−23^	218	1.16 × 10^7^
	hsa-miR-200b-3p	114	1.36 × 10^−22^	151	1.40 × 10^10^
	hsa-miR-107	160	2.30 × 10^−26^	244	2.33 × 10^6^
	hsa-miR-30a-5p	116	5.87 × 10^−24^	174	8.29 × 10^7^
	hsa-miR-141-3p	6	5.83 × 10^8^	103	1.41 × 10^−3^
	hsa-miR-195-5p	107	8.01 × 10^−23^	194	7.92 × 10^6^
	hsa-miR-148a-3p	41	6.11 × 10^−4^	126	1.75 × 10^10^
	hsa-miR-218-5p	69	1.62 × 10^−13^	137	6.68 × 10^9^
	hsa-miR-429	123	2.07 × 10^−23^	120	1.59 × 10^12^
	hsa-miR-128-3p	186	1.15 × 10^−25^	200	4.42 × 10^6^
	hsa-miR-214-3p	60	3.34 × 10^−12^	93	1.85 × 10^−2^
	hsa-miR-24-3p	98	2.58 × 10^−21^	159	6.91× 10^8^
**DE miRNAs from GSE77296**	**miRNA symbol**	**NSR value**	** *p* ** **-value of NSR**	**TFP value**	** *p* ** **-value of TFP**
	hsa-miR-146a-5p	956	4.66 × 10^6^	259	1.95 × 10^−3^
	hsa-miR-200b-3p	114	2.16 × 10^10^	151	3.71 × 10^−2^

## Data Availability

Publicly available datasets (GSE96854, GSE98502, GSE77296 and PXD021292) were analyzed in this study. The first three datasets can be found in the Gene Expression Omnibus database (https://www.ncbi.nlm.nih.gov/geo/ (accessed on 2 January 2023), and the last dataset can be found in the ProteomeXchange Consortium (http://proteomecentral.proteomexchange.org (accessed on 2 January 2023)). The data generated in this study are available in this article.

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
