# Peer review of "Bioinformatics Prediction for Network-Based Integrative Multi-Omics Expression Data Analysis in Hirschsprung Disease"

_biomolecules, 2024, doi:10.3390/biom14020164_

Round 1

Reviewer 1 Report

Comments and Suggestions for Authors

Comments to Author:

I have carefully reviewed the manuscript entitled " Bioinformatics prediction for network-based integrative multi-omics expression data analysis in Hirschsprung Disease" submitted to biomolecules. I appreciate the opportunity to evaluate this research and provide feedback.

The specific amendments are as follows:

1) The writing/language can be improved, there are numerous grammar issues in the manuscript. For example, in line 91, “Data collection and preprocessing Expression data” should be corrected as “Collecting and preprocessing expression data”. Please carefully check the entire text from front to back and correct language problems.

2) Are there any similar articles in this area before? What is the current research bringing in addition to other ones in this field? Some more references should be added in this part.

3) On the one hand, Figure 1 is not clear; on the other hand, readers cannot intuitively find the six steps claimed by the author in Figure 1.

4) There are some risks in similar analysis to Figure 2: Although a gene is a shared differential gene, it is possible that its up- and down-regulation trends are different between two or more groups. Please mark the numbers of up- and down-regulated differentially expressed genes in the Venn diagram respectively.

5) The author conducted a more in-depth discussion of the disease from the perspective of integrated transcriptomics, but in the results we did not see an clarification or depiction of the molecular mechanism (such as the display of molecular mechanism diagrams). This is not easy to understand for a bioinformatics type research paper.

I think that the manuscript would be better if authors could think about the above questions.

Comments on the Quality of English Language

Comments to Author:

I have carefully reviewed the manuscript entitled " Bioinformatics prediction for network-based integrative multi-omics expression data analysis in Hirschsprung Disease" submitted to biomolecules. I appreciate the opportunity to evaluate this research and provide feedback.

The specific amendments are as follows:

1) The writing/language can be improved, there are numerous grammar issues in the manuscript. For example, in line 91, “Data collection and preprocessing Expression data” should be corrected as “Collecting and preprocessing expression data”. Please carefully check the entire text from front to back and correct language problems.

2) Are there any similar articles in this area before? What is the current research bringing in addition to other ones in this field? Some more references should be added in this part.

3) On the one hand, Figure 1 is not clear; on the other hand, readers cannot intuitively find the six steps claimed by the author in Figure 1.

4) There are some risks in similar analysis to Figure 2: Although a gene is a shared differential gene, it is possible that its up- and down-regulation trends are different between two or more groups. Please mark the numbers of up- and down-regulated differentially expressed genes in the Venn diagram respectively.

5) The author conducted a more in-depth discussion of the disease from the perspective of integrated transcriptomics, but in the results we did not see an clarification or depiction of the molecular mechanism (such as the display of molecular mechanism diagrams). This is not easy to understand for a bioinformatics type research paper.

I think that the manuscript would be better if authors could think about the above questions.

Reviewer 2 Report

Comments and Suggestions for Authors

Dear authors,

in this MS you present a framework integrating mRNA and miRNA data to identify potential biomarkers & pathways/terms of interest. To do su you integrate databases of various sources and a stringent workflow. As application you use Hirschsprung's disease. I think this is a very nice work.

It is also well written, except some "fill words", for example in line 111 the word "initially". I suggest scanning the MS for such words and removing them. It makes reading a bit crispier.

However, I would like to see following points in the paper:

First, why did you use the somewhat arbitrary logFC threshold of 0.5 resp. 1.2? It would be nice to add a short explanation. Do the results change a lot if this threshold is not used, and only genes with an FDR<0.05 are used?

Second, you use GEO2R as a starting point of your analysis. The data in GEO are to my knowldedge not subject to quality control. In my experience studies contain outliers and technical failes that should be excluded. How did you ensure the proper assay quality (e.g. by PCA detection for outliers, calculation of NUSE, RLE etc)?

Third, your data contain several different technologies. How did you ensure that technology was not a confounding factor? This should be shown f.e. in a PCA plot or similar.

I think it is a very nice paper with potential and am looking forward to see it published.

Comments on the Quality of English Language

The quality of English is fine, except maybe for some superfluous fill words.

Reviewer 3 Report

Comments and Suggestions for Authors

The authors conducted a computational study to identify genes (protein-coding and miRNA-coding) associated with Hirschsprung Disease, through the combined analysis of differentially expressed genes, protein-protein and miRNA-target interaction networks.

I consider that this work presents new results that may be useful for researchers studying this rare disease, and therefore it deserves to be published. 

Before being acceptable for publication, the text should be revised to improve the quality of the english language used. A few methodological informations should also be altered or corrected to clarify the description of the work done:

1.  on line 134, the thresholds for DE genes use FDR and logFC values. But for DE proteins the threshold are based on p-values (it is not clear if these p-values were FDR corrected of not) and Fold Change (without the log transformation). It would be more clear if the same measures were used in both cases. Moreover, if the Fold change is used without log transformation, it is not symmetric (an upregulation with 1.2 fold change is equivalent to an downregulation of 1/1.2=0.83). Please check that the correct threshold values are being reported.

2.  in table 1, some of the reported properties have not a clear meaning. What is "centralization betweenness" and "centralization degree"? Betweenness and Degree are centralities used to characterize nodes in the network, so, there is not a single value for a complete network. The values presented are the average for all nodes in the network?

3. in lines 370-372, you report the p-value of an hypergeometric test, and state that you assumed an universe of 20000 human genes. As the overlap resulted from the clustering analysis on the ppi network, the correct universe should be the number of protein nodes in the used ppi network (only those proteins could possibly end up in your overlap of interest).

4.  A similar problem occurs in lines 453-454, when a universe of 2300 miRNAs is assumed. Again, there is no need to make assumptions, the miRNAs in the overlap must be part of the miRNAs contained in the databases of MTI used, so the number of unique miRNAs in the MTI databases used in the work should be used as universe.

Comments on the Quality of English Language

Before being acceptable for publication, the text should be revised to improve the quality of the english language used.

Round 2

Reviewer 3 Report

Comments and Suggestions for Authors

The authors have addressed all my comments.